# Statistical precursor signals for Dansgaard-Oeschger cooling transitions

Takahito Mitsui[1,2] and Niklas Boers[1,2,3]

[1]Earth System Modelling, School of Engineering & Design, Technical University of Munich, Germany
[2]Potsdam Institute for Climate Impact Research (PIK), Member of the Leibniz Association, P.O. Box 6012 03, D-14412 Potsdam Germany
[3]Department of Mathematics and Global Systems Institute, University of Exeter, UK

**Correspondence:** Takahito Mitsui (takahito321@gmail.com)

**Abstract.**

Given growing concerns about climate tipping points and their risks, it is important to investigate the capability of identifying robust precursor signals for the associated transitions. In general, the variance and short-lag autocorrelations of the fluctuations increase in a stochastically forced system approaching a critical or bifurcation-induced transition, making them theoretically suitable indicators to warn of such transitions. Paleoclimate records provide useful testbeds if such a warning of a forthcoming transition could work in practice. The Dansgaard-Oeschger (DO) events are characterized by millennial-scale abrupt climate changes during the glacial period, manifesting most clearly as abrupt temperature shifts in the North Atlantic region. Some previous studies find such statistical precursor signals for the DO warming transitions. On the other hand, statistical precursor signals for the abrupt DO cooling transitions have not been identified. Analyzing Greenland ice core records, we find robust and statistically significant precursor signals of DO cooling transitions in most of the interstadials longer than roughly 1500 years, but not in the shorter interstadials. The origin of the statistical precursor signals is mainly related to so-called rebound events, humps in the temperature observed at the end of interstadial, some decades to centuries prior to the actual transition. We discuss several dynamical mechanisms that give rise to such rebound events and statistical precursor signals.

## 1 Introduction

A tipping point is a critical threshold beyond which a system reorganizes, often abruptly and/or irreversibly (IPCC, 2023). Once a tipping point is passed, a system can abruptly transition to an alternative stable or oscillatory state (Boers et al., 2022). Empirical and modelling evidence suggests that some components of the Earth system might indeed exhibit tipping behavior, which poses arguably one of the greatest potential risks in the context of ongoing anthropogenic global warming (Armstrong McKay et al., 2022; Boers et al., 2022). Paleoclimate evidence supports that abrupt climate changes due to crossing tipping points actually occurred in the past (Dakos et al., 2008; Brovkin et al., 2021; Boers et al., 2022). The Dansgaard-Oechger events (Dansgaard et al., 1993) are one of such past abrupt climate changes during the last glacial period, and the focus of this study.

Tipping point behavior is mathematically classified into three different types (Ashwin et al., 2012). 1. Bifurcation-induced tipping is an abrupt or qualitative change of a system owing to a bifurcation of a stable state (more generally a quasi-static attractor); 2. Noise-induced tipping is an escape from a neighborhood of a quasi-static attractor caused by the action of noisy fluctuations (Ditlevsen and Johnsen, 2010); and 3. Rate-induced tipping occurs when a system fails to track a continuously changing a quasi-static attractor (Ashwin et al., 2012; Wieczorek et al., 2023; O'Sullivan et al., 2023). In real world systems, tipping behaviors often result from a combination of several of the above (Ashwin et al., 2012).

The theory of critical slowing down (CSD) provides a framework to anticipate critical (or bifurcation-induced) transitions (Carpenter and Brock, 2006; Scheffer et al., 2009; Kuehn, 2013; Boers, 2018, 2021; Boers and Rypdal, 2021; Boers et al., 2022; Bury et al., 2020). The framework is based on the fact that the stability of a stable state is gradually lost as the system approaches the bifurcation point. Theoretically, the variance of the fluctuations around the fixed point diverges and the autocorrelation with a sufficiently small lag increases toward 1 at the critical point of a codimension-1 bifurcation (Boers et al., 2022; Scheffer et al., 2009; Bury et al., 2020), where the co-dimension-1 bifurcations are, in simple terms, the bifurcations that can be typically encountered by the change of a single control parameter (Thompson and Sieber, 2011). Thus, the changes in CSD indicators such as the increase of the variance as well as the autocorrelation can be seen as *statistical precursor signals* (SPS) of critical transitions. See Dakos et al. (2012) as well as Boers et al. (2022) for various methods and CSD indicators for anticipating critical transitions.

Dansgaard-Oeschger (DO) events are millennial-scale abrupt climate transitions during glacial intervals (Dansgaard et al., 1993). They are most clearly imprinted in the $\delta^{18}$O and calcium ion concentration [Ca$^{2+}$] records from the Greenland ice cores (Fig. 1) (Rasmussen et al., 2014; Seierstad et al., 2014). The $\delta^{18}$O and [Ca$^{2+}$] are interpreted as proxies for site temperature and atmospheric circulation changes, respectively. DO warmings occur typically within a few decades and are followed by gradual cooling during relatively warm glacial states termed interstadials, before a rapid return back to cold states referred to as stadials. The amplitude of the abrupt warming transitions ranges from 5 to 16.5°C (Kindler et al. (2014) and references therein). The Greenland temperatures change concurrently with the North Atlantic temperatures (Bond et al., 1993; Martrat et al., 2004), atmospheric circulation patterns (Yiou et al., 1997), sea-water salinity (Dokken et al., 2013), sea-ice cover (Sadatzki et al., 2019), as well as the Atlantic Meridional Overturning Circulation (AMOC), as inferred from indices such as Pa/Th and $\delta^{13}$C (Henry et al., 2016). The combined proxy evidence suggests that the DO events arise from interactions among these components (Menviel et al., 2020; Boers et al., 2018). The prevailing view is that the mode switching of the AMOC plays a principal role in generating DO events (Broecker et al., 1985; Rahmstorf, 2002), but it remains debated whether the inferred AMOC changes are a driver of DO events or a response to the changes in the atmosphere-ocean-sea ice system in the North Atlantic, Nordic Seas and Arctic (Li and Born, 2019; Dokken et al., 2013). Recently, an increasing number of comprehensive models succeed to simulate DO-like self-sustained oscillations, suggesting that DO events can arise spontaneously from complex interactions between the AMOC, ocean stratification/convection, atmosphere and sea ice (Peltier and Vettoretti, 2014; Vettoretti et al., 2022; Brown and Galbraith, 2016; Klockmann et al., 2020; Zhang et al., 2021; Kuniyoshi et al., 2022; Malmierca-Vallet and Sime, 2023).

The DO events are considered the archetype of climate tipping behavior (Boers et al., 2022; Brovkin et al., 2021). Early works found an SPS based on autocorrelation for one specific DO warming, the onset of Bølling–Allerød interstadial (Dakos et al., 2008). In following works, the existence of SPS for DO warmings was questioned considering that DO warmings are noise-induced rather than bifurcation-induced (Ditlevsen and Johnsen, 2010; Lenton et al., 2012). However a couple of later studies detected SPS for several DO warmings either by ensemble averaging of CSD indicators for many events (Cimatoribus et al., 2013) or by using Wavelet-transform techniques focusing on a specific frequency band (Rypdal, 2016; Boers, 2018). On the other hand, it has so far not been shown whether DO coolings are preceded by characteristic CSD-based precursor signals as well.

Recent studies have inferred that the AMOC is currently at its weakest in at least a millennium (Rahmstorf et al., 2015; Caesar et al., 2018) (see also Kilbourne et al. (2022) for possible uncertainties). The declining AMOC trend is projected to continue in the coming century, although the projections of the AMOC strength in the next hundred years are model-dependent (Masson-Delmotte et al., 2021). Furthermore, the studies applying CSD indicators to observed AMOC fingerprints (Boers, 2021; Ben-Yami et al., 2023; Ditlevsen and Ditlevsen, 2023) as well as a long-term reconstruction of the Atlantic multidecadal variability (Michel et al., 2022) suggest that the AMOC's stability may have declined and the AMOC might be approaching a dangerous tipping point. In this context, it is important to investigate if CSD-based precursor signals can be detected for the DO cooling transitions as well, supposing that the DO events reflect past AMOC changes. Though, of course, predictability of past events does not necessarily imply any predictability in the future, especially given that the recent AMOC weakening is presumably driven by global warming and is thus from a mechanistic point of view different from past AMOC weakenings in the glacial period.

In this study we report SPS for DO cooling transitions recorded in $\delta^{18}$O and $\log_{10}[\text{Ca}^{2+}]$ (Seierstad et al., 2014; Rasmussen et al., 2014) from three Greenland ice cores: NGRIP, GRIP and GISP2 (see Fig. 1 for NGRIP). The important source of observed SPS stems from so-called rebound event, humps in the temperature proxy occurring at the end of interstadials, some decades to centuries prior to the transition (Capron et al., 2010). When CSD indicators such as variance or lag-1 autocorrelation are used for anticipating a tipping point, we conventionally assume that a system gradually approaches a bifurcation point. However, if DO cycles are spontaneous oscillations as suggested in some GCMs (see above), in a strict sense there might not be any bifurcation occurring around the timings of the abrupt transition in the DO cycles. Nevertheless, with conceptual models of DO cycles, we demonstrate that CSD indicators may show precursor signals for abrupt transitions due to particular dynamics near a critical point or a critical manifold.

The reminder of this paper is organized as follows. In Section 2, the data and method are described. In Section 3, we identify robust and statistically significant SPS for several DO cooling transitions following interstadials with sufficient data length, and show that rebound events prior to cooling transition are the source of observed statistical precursor signals. In Section 4 we discuss the results by using conceptual models. A summary is given in Section 5.

## 2 Data and methods

 ### 2.1 Greenland ice core records

We explore CSD-based precursor signals for DO cooling transitions recorded in $\delta^{18}O$ and $\log_{10}[Ca^{2+}]$ (Seierstad et al., 2014; Rasmussen et al., 2014) from three Greenland Ice Cores: NGRIP, GRIP and GISP2 (see Fig. 1 for NGRIP). Multiple records are used for a robust assessment because each has regional fluctuations as well as proxy- and ice-core-dependent uncertainties. The six records have been synchronized and are given at 20-year resolution (Seierstad et al., 2014; Rasmussen et al., 2014). They continuously span the last 104 kyr b2k (kiloyears before 2000 CE), beyond which only NGRIP $\delta^{18}O$ is available up to a part of the Eemian interglacial. In addition, we use a version of the NGRIP $\delta^{18}O$ and dust records at 5-cm depth resolution (epica community members, 2004; Gkinis et al., 2014; Ruth et al., 2003) in order to check the dependence of results on temporal resolutions, with the caveat that these high-resolution records span only the last 60 kyr.

We follow the classification of interstadials and stadials and associated timings of DO warming and cooling transitions by Rasmussen et al. (2014), where Greenland interstadials (stadials) are labelled with 'GI' ('GS') with few exceptions below. A *rebound event* is an abrupt warming often observed before an interstadial abruptly ends (Capron et al., 2010) (arrows in Figs. 1, 2 and 3). Generally a long interstadial accompanies a long rebound event (their durations are correlated with $R^2 = 0.95$, Capron et al. (2010)). In Capron et al. (2010) and Rasmussen et al. (2014), GI-14 and subsequent GI-13 are seen as one long interstadial with GI-13 consdered to be a strongly expressed rebound event ending GI-14 because the changes in $\delta^{18}O$ and $\log_{10}[Ca^{2+}]$ during the quasi-stadial GS-14 do not reach the base-line levels of adjacent stadials. Similarly GI-23.1 and GI-22 are also seen as one long interstadial, and GI-22 is regarded as a rebound event (and GS-23.1 as quasi-stadial) (Capron et al., 2010; Rasmussen et al., 2014). GI-20a is also recognized as a rebound event in Rasmussen et al. (2014). Given that the rebound events are warmings following a colder spell during interstadial conditions that does not reach the stadial levels (Rasmussen et al., 2014), we regard the following nine epochs as rebound-type events: GI-8a, the hump at the end of GI-11 (42240–~42500 yr b2k), GI-12a, GI-13, the hump at the end of GI-16.1 (56500–~56900 yr b2k), GI-20a, GI-21.1c-b-a (two warming transitions), GI-22 and GI-25a. When we examine the effect of rebound events on our results, we exclude the entire parts including the cold spells prior to the rebound events.

The start (warming) and end (cooling) of each DO event are identified in 20-yr resolution based on both $\delta^{18}O$ and $[Ca^{2+}]$ in Rasmussen et al. (2014). The estimated uncertainty of event timing varies from event to event. We remove the $2\sigma$ uncertainty range of the event timing (40 to 400 years) estimated in Rasmussen et al. (2014) from our calculation of CSD indicators. It effectively excludes parts of the transitions themselves from the calculation of the CSD indicators. Since the calculation of CSD indicators requires a minimum length of data points, we mainly focus on interstadials longer than 1000 yr after removing $2\sigma$ uncertainty ranges of the transition timings, using 20-yr resolution data (Table S1). This results in 12 DO interstadials to be investigated (Fig. 1, gray shades). We deal with the interstadials shorter than 1000 yr but longer than 300 yr using high resolution records in Section 3.2.

## 2.2 Statistical indicators of critical slowing down

Based on the theory of critical slowing down (CSD), we posit that the stability of a dynamical system perturbed by noise is gradually lost as the system approaches a bifurcation point (Boers et al., 2022; Scheffer et al., 2009; Bury et al., 2020). For the fold bifurcation (also known as the saddle-node bifurcation), the variance of the fluctuations around a local stable state diverges and the autocorrelation function of the fluctuations increases toward 1 at any lag $\tau$. The same is true for the transcritical as well as the pitchfork bifurcation (Bury et al., 2020). For the Hopf bifurcation, the variance increases, but the autocorrelation function of the form $C(\tau) = e^{\nu|\tau|}\cos\omega\tau$ may increases or decreases depending on $\tau$, where $\nu(\le 0)$ and $\pm\omega i$ are the real and imaginary parts of the complex eigenvalues of the Jacobian matrix of the local linearized system (Bury et al., 2020). Nevertheless the autocorrelation function $C(\tau)$ increases for sufficiently small $\tau$. For discrete time series, we follow previous studies and calculate the lag-1 autocorrelation corresponding to a minimal $\tau$. These characteristics can be used to anticipate abrupt transitions cause by codimension-one bifurcations. Promisingly previous studies show that these CSD indicators and related measures can indeed anticipate simulated AMOC collapses (Boulton et al., 2014; Klus et al., 2018; Livina and Lenton, 2007; Held and Kleinen, 2004).

Prior to calculating CSD indicators, we estimate the local stable state by using a local regression method called the locally weighted scatterplot smoothing (LOESS) (Cleveland et al., 1992; Dakos et al., 2012). In this approach, the time series is seen as a scatter plot and fitted locally by a polynomial function, giving more weights to points near the point whose response is being estimated and less weights to points further away. Here the polynomial degree is set to 1, i.e., the smoothing is performed with the local linear fit. Nevertheless, the LOESS provides nonlinear smoothed curves. The smoothing span parameter $\alpha$ that defines the fraction of data points involved in the local regression is set to 50% of each interstadial length in a demonstration case, but we examine the dependence of results on $\alpha$ over the range 20–70%. The difference between the record and the smoothed one gives the residual fluctuations. The CSD indicators, i.e., variance and lag-1 autocorrelation, are calculated for the residuals over a rolling window. The size $W$ of this rolling window is set to 50% of each interstadial length in the demonstration case. To test robustness this is changed over the range 20–60%.

The statistical significance of precursor signals of critical transitions, in terms of positive trends of CSD indicators, is assessed by hypothesis testing (Theiler et al., 1992; Dakos et al., 2012; Rypdal, 2016; Boers, 2018). We consider as null model a stationary stochastic process with preserved variance and autocorrelation. The $n$ surrogate data are prepared form the original residual series by the phase-randomization method, thus preserving the variance and autocorrelation function of the original time series via the Wiener-Khinchin theorem. Here we take $n = 1000$. The linear trend ($a_o$) of the CSD indicator for the original time series and the linear trends ($a_s$) of CSD indicators for the surrogate data are calculated. We consider the precursor signal of the original time series statistically significant at 5% level if the probability of $a_s > a_o$ ($p$-value) is less than 0.05. The significance level of 0.05 is conservative given that some works analyzing ecological or paleo-data adopt the significance level of 0.1 (Dakos et al., 2012; Thomas et al., 2015).

## 3 Results

### 3.1 Characteristic precursor signals of DO coolings

As CSD indicators we consider the variance and lag-1 autocorrelation, calculated in rolling windows across each interstadial. The 12 interstadials longer than 1000 yr are magnified in Figs. 2 and 3 (top rows, blue) for the NGRIP $\delta^{18}$O record. See Figs. S1–S10 for the other records. For each interstadial, the nonlinear trend is estimated using LOESS smoothing (Figs. 2 and 3, top row, red). In this case the smoothing span $\alpha$ that defines the fraction of data points involved in the local regression is set to 50% of each interstadial length. Gaussian kernel smoothing gives similar results. The difference between the record and

the nonlinear trend gives the approximately stationary residual fluctuations (second row). The CSD indicators are calculated from the residual series over a rolling window. In Figs. 2 and 3 the rolling window size $W$ is set to 50% of each interstadial length (a default value in Dakos et al. (2008)). The smoothing span $\alpha$ and the rolling window size $W$ are taken as fractions of individual interstadial length because time scales of local fluctuations (such as the duration of rebound events) change with the entire duration of interstadial. We examine the dependence of the results on $\alpha$ and $W$ as part of our robustness tests.

The variance is plotted in the third row of Figs. 2 and 3. Positive trends in the variance are observed for 9 out of 12 interstadials; the individual trends are statistically significant in 6 out of 12 cases ($p < 0.05$), based on a null model assuming the same overall variance and autocorrelation, constructed by producing surrogates with randomized Fourier phases. The lag-1 autocorrelation is also plotted for the same data in the bottom row. Positive trends in lag-1 autocorrelation are observed for 10 out of 12 interstadials, but are statistically significant only in 2 cases ($p < 0.05$). Just a positive trend without significance

cannot be considered a reliable SPS, but if one indicator has a significantly positive trend, the other indicator with consistently positive trend may at least support the conclusion (e.g., GI-19.2 and GI-14-13 in Fig. 3). In several cases (GI-24.2, 21.1, 16.1, 14-13 and 12), the lag-1 autocorrelation first decreases and then increases. The initial decreases, harming monotonic increases of CSD indicators, might reflect a memory of the preceding DO warming transition. On the other hand, the drastic increases in both indicators near the end of the interstadials reflect the rebound events (arrows in Figs. 2 and 3). We obtain similar results

for the other ice core records (Figs. S1–S10). While we observe a number of positive trends for all the records, the number of detected statistically significant trends depends on the record and CSD indicator (Table S2).

We check robustness of our results against changing smoothing span $\alpha$ and rolling window size $W$ (Dakos et al., 2012). We calculate the $p$-value for the trend of each indicator changing the smoothing span between 20–70% of interstadial length (in steps of 10%) and the rolling window size between 20–60% (also in 10% steps), respectively. This yields a $6 \times 5$ matrix

for the $p$-values: Example results, for GI-25 and $\delta^{18}$O, are shown in Figs. 4a (variance) and 4b (lag-1 autocorrelation). The cross mark (x) indicates significant positive trends ($p < 0.05$) and the small open circle (o) indicates positive trend that are significant at 10% level but not at 5% level. Full results for the 12 interstadials, 6 records, and two CSD indicators are shown in Figs. S11–S22. We consider positive trends in CSD indicators, i.e. the SPS of the transition, to be overall robust if we obtain significant positive trends ($p < 0.05$) for more than half ($> 15$) of the 30 parameter sets.

The robustness analysis is performed for all the long interstadials of the 6 records and the two CSD indicators (Fig. 4d). Among the 12 interstadials, we find at least one robust SPS for 8 interstadials (GI-25, 23.1, 21.1, 20, 19.2, 14, 12 and 8) and

multiple robust SPS for 6 (GI-25, 23.1, 21.1, 14, 12 and 8). If the data series is a stationary stochastic process, the probability of spuriously observing a robust SPS is estimated to be 5% (Appendix A). In this case, the probability of detecting more than 2 robust SPS from 12 independent stationary samples (i.e., from each row in Fig. 4d) by chance is only ~2%. Thus, the risk of obtaining the results by chance is quite low. For each interstadial, the detection of robust SPS depends on the proxy and core. This is possibly because different proxies from different cores are contaminated by different types and magnitudes of noise (e.g., $\delta^{18}O$ may record local fluctuations of temperatures and $\log_{10}[Ca^{2+}]$ turbulent fluctuations of local wind circulations). Robust SPS are observed for most interstadials longer than roughly 1500 yr (GI-25, 23.1, 21.1, 20, 19.2, 14, 12 and 8 except GI-1) but not for the other interstadials, shorter than roughly 1500 yr (compare Figs. 4c and 4d).

## 3.2   Further sensitivity analyses

We examine how much the rebound events affect the detection of CSD-based SPS. For this purpose CSD indicators are again calculated excluding the rebound events and their preceding cold spells (see Section 2.1). While 8 interstadials (GI-25, 23.1, 21.1, 20, 16, 14, 12 and 8) exhibit robust SPS with the rebound events included, only four interstadials (GI-23.1, 14, 12 and 8) exhibit robust SPS without the rebound events (Fig. S23). The rebound events should hence be considered important, sometimes indispensable, sources for SPS of DO coolings.

We also examine the dependence of the results on the time resolution of the data. Here we use a high-resolution (5-cm depth) $\delta^{18}O$ record (**?**Gkinis et al., 2014) and a dust record (Ruth et al., 2003) from the NGRIP over the last 60 kyr. Since the data in these records are non-uniform in time, they are linearly resampled every 5 yr before calculating CSD indicators. We focus on 11 interstadials longer than 300 yr in order to have enough data points. For the dust record, three interstadials (GI-15, 8 and 7) are excluded from the analysis because the original data has long parts of missing values. The CSD indicators, calculated with a smoothing span of $\alpha = 50\%$ and rolling windows with $W = 50\%$, are shown in Figs. S24–S27. Through the robustness analyses with respect to $\alpha$ and $W$, we find at least one robust SPS for 3 out of 11 interstadials (Fig. S28). The robust SPS for GI-14-13 and GI-12 from the high-resolution records are consistent with those from the 20-yr resolution records. Moreover for GI-1, the high-resolution $\delta^{18}O$ record exhibits a robust SPS in terms of lag-1 autocorrelation, although the 20-yr resolution record does not. Robust SPS have not been observed again for interstadials shorter than roughly 1500 yr (Figs S28 and 4).

## 4   Possible dynamical mechanisms

We detected robust precursor signals of DO cooling transitions for most interstadials longer than roughly 1500 yr, but not for shorter interstadials. The results suggest that long interstadials, the existence of rebound events, and the presence of SPS for the DO cooling transitions are all related (except for GI-19.2, which has no noticeable rebound event). These aspects may be related to generic properties of nonlinear dynamical systems. On the basis of conceptual mathematical models, we discuss four possible dynamical mechanisms leading to the precursor signals of DO cooling transitions. In three of four mechanisms, oscillations like the rebound events can systematically arise prior to the abrupt cooling transitions. These modelling results

justify the inclusion of the rebound events in the search for precursor signals presented above. Unless otherwise mentioned, details on model parameters as well as conducted hysteresis experiments are given in Appendix B.

1. The fold bifurcation mechanism. Since the pioneering work by Stommel (1961), the AMOC is considered to exhibit bistability depending on the background condition (Rahmstorf, 2002). The bistability of the AMOC strength $x$ may be conceptually modelled by a double-fold bifurcation model: $\dot{x} = f(x) + \mu + \xi(t)$, where $f(x)$ has two fold points like $x - x^3$ and $|x|(1-x)$. Here we take the quadratic from $f(x) = |x|(1-x)$, but the following arguments are qualitatively the same for $x - x^3$. The parameter $\mu$ represents the surface salinity flux (i.e. negative freshwater flux), and $\xi(t)$ denotes white Gaussian noise. The unperturbed model for $\xi(t) = 0$ has equilibria on an S-shaped curve: $f(x) + \mu = 0$ (Fig. 5a, green). The state $x(t)$ initially on the upper stable branch jumps down to the lower stable branch as $\mu$ decreases across the fold bifurcation point at $\mu = -0.25$. The variance and the autocorrelation of the local fluctuations (i.e. CSD indicators) increase as $\mu$ slowly approaches the fold bifurcation point since the restoring rate toward the stable state decreases, as shown in Fig. 6a (Scheffer et al., 2009; Boers et al., 2022).

2. Stochastic slow-fast oscillation mechanism. The FitzHugh-Nagumo (FHN) system is a prototypical model for slow-fast oscillations and excitability (FitzHugh, 1961; Nagumo et al., 1962). It is often invoked for conceptual models of DO oscillations (Rial and Yang, 2007; Kwasniok, 2013; Roberts and Saha, 2017; Mitsui and Crucifix, 2017; Lohmann and Ditlevsen, 2019; Riechers et al., 2022; Vettoretti et al., 2022). An FHN-type model of DO oscillations can be obtained by introducing a slow variable $y$ into the fold bifurcation model: $\dot{x} = \frac{1}{\tau_x}(|x|(1-x) + y + \mu) + \xi(t)$, $\dot{y} = \frac{1}{\tau_y}(-x - y)$, where $\tau_x$ and $\tau_y$ are time-scale parameters ($\tau_x \ll \tau_y$). Invoking the salt-oscillator hypothesis for DO oscillations suggested by the comprehensive climate model simulations that are successful in reproducing DO cycles (Vettoretti and Peltier, 2018), we may interpret $y$ as the surface mixed layer salinity in the northern North Atlantic and Labrador sea, which gradually decreases (increases) when the AMOC intensity $x$ is strong (weak).

Here we consider the case that the system is excitable. For example for $\mu = 0.26$, the unperturbed system has a stable equilibrium near the upper fold point of the S-shaped critical manifold, $\{(x, y) \in \mathbb{R}^2 \mid y = -|x|(1-x) - \mu\}$ (Fig. 5c, green), but the dynamical noise $\xi(t)$ enables the escape from the barely stable equilibrium and sustains stochastic oscillations (Figs. 5b and 5c, blue). Due to the time-scale separation ($\tau_x \ll \tau_y$), the oscillations occur along the attracting parts of the critical manifold (Fig. 5c). Because $y$ is much slower than $x$, the dynamics of $x$ is similar to the dynamics of the fold bifurcation model with slowly changing $y$. As a result, SPS can be effectively observed near the upper fold point of the critical manifold (Fig. S29). However, this example is not rigorous bifurcation-induced tipping. In the example of an excitable system (Figs. 5b and 5c), the underlying system always has a weakly stable fixed point, and no true bifurcation leading to critical slowing down occurs. In fact, the actual tipping in this case is noise-induced. However, we can effectively observe the SPS in CSD indicators in this case as well, since the system would in each cyclic iteration move from more stable to less stable conditions until it finally tips to initiate the next cycle; and this partial decrease in stability is imprinted in the CSD indicators (Fig. S29). The increase of the variance prior to the transitions in the FHN model is reported also in Meisel and Kuehn (2012). Since the unperturbed system has an equilibrium near the upper fold point, the motion is stagnant near the fold point. This provides favorable conditions for observing SPS. The state jumps from the upper branch of the critical manifold to its lower blanch often occur after an

upward jump induced by noise. These upward jumps resemble the rebound events prior to DO cooling transitions. The overall phenomenon is the same in the self-sustained oscillation regime of the FHN model, as long as the equilibrium locates near the upper fold point of the critical manifold ($\mu \simeq 0.25$).

3. Hopf bifurcation mechanism. In contrast to the fold bifurcation, the Hopf bifurcation manifests oscillatory instability (Strogatz, 2018). In several ocean box models, the thermohaline circulation is destabilized via a Hopf bifurcation (Alkhayuon et al., 2019; Lucarini and Stone, 2005; Abshagen and Timmermann, 2004; Sakai and Peltier, 1999). It is also considered a potential generating mechanism of DO oscillations in a low-order coupled climate model (Timmermann et al., 2003) and in a comprehensive climate model (Peltier and Vettoretti, 2014). Assume that the parameter $\mu$ decreases slowly in the FHN-type model (Figs. 5d and 5e). The underlying dynamics changes from the stable equilibrium to the limit-cycle oscillations at the Hopf bifurcation point $\mu = \mu_{\mathrm{Hopf}} \equiv (1 - \tau_x/\tau_y)^2/4$ (Appendix B). If stochastic forcing is added to the system, noise-induced small oscillations can appear prior to the onset of the limit-cycle oscillations (Fig. 5d and 5e). The precursor oscillations resemble rebound events, while their shape depends on the noise as well as the change rate of $\mu$. Again SPS can be observed near the Hopf bifurcation point (Fig. S30) (Bury et al., 2020; Meisel and Kuehn, 2012; Boers et al., 2022). The small oscillations prior to downward transitions, like the DO rebound events, do not appear if the system goes deeply into the self-sustained oscillation regime away from the Hopf-bifurcation point ($\mu < \mu_{\mathrm{Hopf}} \approx 0.245$ in Fig. 5d).

4. Mixed-mode oscillation mechanism. Mixed-mode oscillations (MMOs) are periodic oscillations consisting of small and large-amplitude oscillations (Koper, 1995; Desroches et al., 2012; Berglund and Landon, 2012). They often arise in systems with one fast variable and two slow variables (Desroches et al., 2012). In this regard, we can extend the above FHN-type model to exhibit MMOs, for example, as follows: $\dot{x} = \frac{1}{\tau_x}(|x|(1-x)+y+\mu)$, $\dot{y} = \frac{1}{\tau_y}(-x-y+k(z-y))$ and $\dot{z} = \frac{1}{\tau_z}(-x-z+k(y-z))$, where $z$ is another slow variable with time scale $\tau_z$ ($\gg \tau_x$) and $k$ is the diffusive-coupling constant between slow variables. We interpret $y$ as the surface salinity in the northern North Atlantic convection region that directly affects the AMOC strength $x$ again, and $z$ as the surface salinity outside the convection region that affects the surface salinity $y$ in the convection region via mixing. This extended FHN-type model is introduced here just for demonstrating that MMOs may appear in a FHN-type model with a minimal dimensional extension. For certain parameter settings (Appendix B), the system has an unstable equilibrium $(x, y, z) = (\sqrt{\mu}, -\sqrt{\mu}, -\sqrt{\mu})$ of saddle-focus type, with one stable direction with a negative real eigenvalue and a two-dimensional unstable manifold with complex eigenvalues with positive real part. The slow-fast oscillations occur along the critical manifold $\{(x, y, z) \in \mathbb{R}^3 \,|\, y = -|x|(1-x) - \mu\}$ (Figs. 5f and 5g). However, due to the saddle-focus equilibrium on the critical manifold, the trajectory is attracted toward the saddle from the direction of the stable manifold (black segment) and repelled from it in a spiralling fashion. The striking point is the systematic occurrence of small-amplitude oscillations prior to the abrupt transition, which also resemble the rebound events prior to the DO cooling transitions. A more realistic time series is obtained if an observation noise is added on $x(t)$ (Fig. S31). Then SPS can be stably observed near the fold point of the critical manifold.

Based on the four types of simple mathematical models, we have proposed four possible dynamical mechanisms for the DO cooling transitions that can manifest statistical precursor signals (SPS): (1) strict critical slowing down due to the approaching of a fold bifurcation, (2) critical slowing down in a wider sense, in stochastic slow-fast oscillations, (3) noise-induced oscil-

lations prior to Hopf bifurcations, or (4) the signal of mixed-mode oscillations. While the details of these mechanisms are different, they are all related to the fold points of the equilibrium curve or the critical manifolds. As a result, the SPS can be
detected by the conventional CSD indicators.

The mechanisms (2), (3) and (4) can generate behavior resembling the rebound events, leading to increases in the classical CSD indicators. In the toy models, rebound event-like behavior is generated when the trajectory passes by an equilibrium point with marginal stability (i.e., the equilibrium has neither strong stability leading to a permanent state nor strong instability leading to short interstadials) (Figs. 5b–5g). In this case, the duration of the modelled interstadial is relatively long in relation
to the marginal stability. In contrast, the absence of equilibrium or the presence of a strongly unstable equilibrium near the fold point of the critical manifold leads to brief interstadials without a rebound event and consequently a lack of SPS. This provides a possible explanation why the rebound events and the robust SPS are simultaneously observed for long interstadials, but not for short interstadials.

Another possible explanation for the lack of SPS for short interstadials is following. The common assumption underlying
CSD theory is that the parameter change is much slower than the system's relaxation time, and the latter is much slower than the correlation time of the noise (Thompson and Sieber, 2011; Ashwin et al., 2012). If this assumption is violated, it is difficult to detect CSD-based SPS (Clements and Ozgul, 2016; van der Bolt et al., 2021). Consider the fold-bifurcation-induced tipping in the Stommel-type model (1) for example (Fig. 6). If the change in the parameter $\mu$ is faster than the system's relaxation time toward the moving stable equilibrium, it is unlikely to detect significant CSD-based SPS (Fig. 6b) because the trajectory
evolves systematically away from the equilibrium and thus cannot feel the flattening of the potential around the equilibrium even at the true bifurcation point.

## 5   Summary and discussion

In this study we have explored statistical precursor signals (SPS), significant increases in critical slowing down (CSD) indicators (variance and lag-1 autocorrelation), for Dansgaard-Oeschger (DO) cooling transitions following interstadials, using six
Greenland ice core records. Among the 12 interstadials longer than 1000 yr, we find at least one robust SPS for 8 interstadials longer than roughly 1500 yr (GI-25, 23.1, 21.1, 20, 19.2, 14, 12 and 8) and multiple robust SPS for 6 of them (GI-25, 23.1, 21.1, 14, 12 and 8) (Fig. 4d). Robust SPS are, however, not observed for interstadials shorter than roughly 1500 yr. One might link the increase in the proxy variance with the tendency of larger climatic fluctuations in colder climates (Ditlevsen et al., 1996), but the increases in the lag-1 autocorrelation cannot generally be explained by it. The analysis removing the rebound
events from the data shows that the rebound events prior to the cooling transitions are important in producing the SPS.

We have proposed four different dynamical mechanisms to explain the observed SPS: (1) strict critical slowing down due to the approaching of a fold bifurcation, (2) critical slowing down in a wider sense, in stochastic slow-fast oscillations, (3) noise-induced oscillations prior to Hopf bifurcations, or (4) the signal of mixed-mode oscillations. In the latter three mechanisms, oscillations like the rebound events can systematically arise prior to the abrupt cooling transitions. These precursor
oscillations are due to marginally-(un)stable equilibria on the critical manifolds that cause long-lived quasi-stable state (like

long interstadials). This can explain why rebound events and SPS are simultaneously observed only for long interstadials and are not observed for short ones. While the SPS for bifurcation-induced tipping events (mechanism 1 and 3) are established, detailed properties of SPS for the stochastic slow-fast oscillations of the excitable system (mechanism 2) and for the mixed-mode oscillations (mechanism 4) remain to be elucidated.

We should mention the assumptions taken in this study, as well as alternative scenarios for the DO cooling transitions. First, the above four dynamical mechanisms assume slow changes in parameters or slow variables which cause bifurcations in the fast subsystem. On the other hand, the rate-induced tipping mechanism has also been invoked for a possible AMOC collapse, where the rate of the change of the external forcing (e.g., freshwater flux or atmospheric $CO_2$ concentration) determines the future AMOC state (Alkhayuon et al., 2019; Lohmann and Ditlevsen, 2021; Ritchie et al., 2023). The lack of observed SPS for the interstadials less than roughly 1500 yr indicates a rate-dependent aspect of the DO cooling transitions. However, a comprehensive investigation of DO cooling transitions from the viewpoint of rate-induced tipping is beyond the scope of this work. Second, a recent study using an ocean general circulation model shows that a rebound-event-type behavior of AMOC is caused by a behavior called the intermediate tipping, due to multiple stable ocean circulation states that exist near but prior to the tipping point leading to a significant AMOC weakening (Lohmann et al., 2023). The intermediate tipping mechanism for rebound events is different from the possible low-dimensional dynamical mechanisms proposed in this study. Further studies are needed to elucidate the dynamical as well as physical origin of DO coolings and associated rebound events.

We have shown that past abrupt DO cooling transitions in the North Atlantic region can be anticipated based on classical CSD indicators if they are preceded by long interstadials. However, it is found to be difficult to anticipate DO cooling events, at least from the 20-yr-resolution ice core Greenland records, if they occur after a short interstadial. If the DO coolings transitions are actually associated with AMOC weakening (see the Introduction), our results may have an implication on the predicted weakening of the AMOC and its possible collapse in the future: the prediction with CSD indicators could be more difficult if the forcing changes fast. There is, however, a caveat to this implication because the past DO cooling transitions are different from the presently inferred AMOC changes. The time resolution (mainly 20 years and additionally 5 years) and the length (mainly >1000 years and additionally >300 years) of the interstadial segment data used in this study are coarser and mostly longer than the annual data used for analyzing AMOC fingerprints during the industrial period (Boers, 2021; Ben-Yami et al., 2023; Ditlevsen and Ditlevsen, 2023) and the last millennium (Michel et al., 2022). More crucially, the revealed predictability of past DO cooling events does not necessarily imply predictability of a potential future AMOC collapse since the recent AMOC weakening, possibly driven by global warming but potentially also part of natural variability, is mechanistically very different from past AMOC weakening in the glacial period.

**Supplement**

The supplement related to this article is available online at: https://github.com/takahito321/Predictability-of-DO-cooling/blob/main/EWS_DO_cooling_SI.pdf

*Code and data availability.* The Greenland ice core records used in this study can be obtained from https://www.iceandclimate.nbi.ku.dk/data/. The R-codes used in this study are available from https://github.com/takahito321/Predictability-of-DO-cooling.git.

## Appendix A: Probability of observing robust precursor signals

The statistical significance of precursor signals of critical transitions, in terms of positive trends of CSD indicators, is assessed by hypothesis testing (Theiler et al., 1992; Dakos et al., 2012; Rypdal, 2016; Boers, 2018). We consider as null model a stationary stochastic process with preserved variance and autocorrelation. The $n$ surrogate data are prepared form the original residual data series by the phase-randomization method, thus preserving the variance and autocorrelation function of the original time series via Wiener-Khinchin theorem. Here we take $n = 1000$. The linear trend ($a_0$) of the CSD indicator for the original residual time series and the linear trends ($a_s$) of CSD indicators for the surrogate data are calculated. We consider the precursor signal of the original series statistically significant at 5% level if the probability of $a_s > a_o$ ($p$-value) is less than 0.05. Thus, if the original data is already a stationary stochastic process (exhibiting no CSD), one should expect spuriously significant results at a probability of 0.05 by definition. In principle this is independent of the smoothing span $\alpha$ as well as the rolling window size $W$ used for calculating CSD indicators. We consider a precursor signal robust if we find significant cases ($p < 0.05$) for more than half ($> 15$) of 30 combinations of $\alpha$ and $W$. Then the probability of observing a robust precursor signal can be shown to be 0.05. In order to check this numerically, we generate 5000 surrogates of the original $\delta^{18}O$ series of interstadial GI-25 and calculate the probability of finding robust precursor signals. The resulting fractions are 0.041 for the variance and 0.039 for the lag-1 autocorrelation, which are close to 0.05. For the case of GI-12, we obtain 0.038 for the variance and 0.047 for the lag-1 autocorrelation, again close to 0.05. These results support that the probability of observing a robust precursor signal is 5% if the data are stationary stochastic processes.

## Appendix B: Details of conceptual models used in Fig. 5

Here we describe specific settings for four conceptual models representing different candidate mechanisms for the DO cooling transitions. Unless otherwise mentioned, stochastic differential equations below are solved with the Euler-Maruyama method with step size of $10^{-3}$.

1. The bistability of the AMOC strength $x$ can be conceptually modelled by a double-fold bifurcation model: $\dot{x} = f(x) + \mu + \xi(t)$, where $f(x)$ has two fold points, here for $f$ one can use either $f(x) = x - x^3$ or $f(x) = |x|(1 - x)$. We take the quadratic function $f(x) = |x|(1 - x)$ that arises in Stommel (1961) model. $\mu$ represents a forcing parameter on the AMOC strength $x$, e.g., salinity forcing on the North Atlantic (i.e., negative freshwater forcing). $\xi(t)$ is white Gaussian noise, e.g. freshwater perturbations or weather forcing. In Fig. 5a, we set $\sqrt{\langle \xi^2 \rangle} = 0.03$, and the initial condition is taken at $x(0) = 1.1$, near the upper stable fixed point of the unperturbed system. The parameter $\mu$ is then slowly decreased from 0.1 to $-0.4$ over the period from $t = 0$ to $t = 500$, to trigger the bifurcation-induced transition.

2. The FitzHugh-Nagumo-type (FHN-type) system is a prototypical model of slow-fast oscillators (FitzHugh, 1961; Nagumo et al., 1962) and often invoked for conceptual models of DO oscillations (Rial and Yang, 2007; Kwasniok, 2013; Roberts and Saha, 2017; Mitsui and Crucifix, 2017; Lohmann and Ditlevsen, 2019; Riechers et al., 2022; Vettoretti et al., 2022). The FHN-type model subjected to dynamical noise can be obtained by introducing a slow variable $y$ into the fold bifurcation model: $\dot{x} = \frac{1}{\tau_x}(|x|(1-x) + y + \mu) + \xi(t)$, $\dot{y} = \frac{1}{\tau_y}(-x - y)$, where $\tau_x$ and $\tau_y$ are time-scale parameters ($\tau_x \ll \tau_y$). Following the salt-oscillator hypothesis of to explain DO cycles (Vettoretti and Peltier, 2018), we may interpret $y$ as the salinity in the polar halocline surface mixed layer, which decreases (increases) when the AMOC is strong (weak). In the case of Fig. 5 we set $\mu = 0.26$, $\tau_x = 0.01$, $\tau_y = 1$ and $\sqrt{\langle \xi^2 \rangle} = 0.3$, and the initial condition is taken at $(x(0), y(0)) = (-0.2, -0.45)$. The $x$-nullcline (critical manifold) of the unperturbed system is $y = -|x|(1-x) - \mu$ (Fig. 5c, green) and the $y$-nullcline is the $y = -x$ (Fig. 5c, magenta dashed). The intersection of the $x$- and $y$-nullclines is the equilibrium point of the unperturbed system $(\sqrt{\mu}, -\sqrt{\mu})$, which is near the fold point of the critical manifold in this parameter setting.

3. For demonstrating the Hopf bifurcation mechanism in Figs. 5d and 5e, the same stochastic FHN-type model is used with $\tau_x = 0.01$, $\tau_y = 1$ and $\sqrt{\langle \xi^2 \rangle} = 0.05$, but here $\mu$ is gradually decreased from 0.3 to 0.2, over a period of 5 time units. For $0.2 < \mu < 0.3$, the system has a unique equilibrium point at $(x, y) = (\sqrt{\mu}, -\sqrt{\mu})$. The Hopf bifurcation of an equilibrium occurs if the complex eigenvalues of the Jacobian matrix at the equilibrium passes the imaginary axis (Strogatz, 2018). The eigenvalues of the Jacobian matrix at this equilibrium are $\lambda_\pm = \frac{1}{2}\{\frac{1-2\sqrt{\mu}}{\tau_x} - \frac{1}{\tau_y} \pm \sqrt{(\frac{1-2\sqrt{\mu}}{\tau_x} - \frac{1}{\tau_y})^2 - \frac{8\sqrt{\mu}}{\tau_x \tau_y}}\}$. These eigenvalues $\lambda_\pm$ are complex conjugates for $\frac{1}{4}(1 + \frac{\tau_x}{\tau_y} - 2\sqrt{\frac{\tau_x}{\tau_y}})^2 < \mu < \frac{1}{4}(1 + \frac{\tau_x}{\tau_y} + 2\sqrt{\frac{\tau_x}{\tau_y}})^2$. In this range of $\mu$, the real part of $\lambda_\pm$ changes from negative to positive at the Hopf bifurcation point: $\mu_{\text{Hopf}} = \frac{1}{4}(1 - \frac{\tau_x}{\tau_y})^2$. For $\tau_x/\tau_y = 0.01$, $\mu_{\text{Hopf}} \approx 0.245$.. The initial condition is taken at the origin.

4. The mixed-mode oscillation model is obtained if the FHN-type model is extended to have multiple interacting slow variables. E.g., $\dot{x} = \frac{1}{\tau_x}(|x|(1-x) + y + \mu)$, $\dot{y} = \frac{1}{\tau_y}(-x - y + k(z - y))$ and $\dot{z} = \frac{1}{\tau_z}(-x - z + k(y - z))$, where $z$ is another slow variable with time scale $\tau_z$ ($\gg \tau_x$) and $k$ is the diffusive coupling constant between slow variables. We interpret $y$ as the surface salinity in the northern North Atlantic convection region, which directly affects the AMOC strength $x$, and $z$ as the surface salinity outside the convection region that affects the surface salinity $y$ in the convection region via mixing. We set $\tau_x = 0.02$, $\tau_y = 2$, $\tau_z = 4$, $\mu = 0.225$ and $k = 0.8$. This system has an unstable equilibrium $(x, y, z) = (\sqrt{\mu}, -\sqrt{\mu}, -\sqrt{\mu})$ of saddle-focus type, with one stable direction with a negative real eigenvalue $-0.67$ and a two-dimensional unstable manifold with two complex conjugate eigenvalues with positive real part $0.94 \pm 4.7i$. The initial condition is taken at $(x(0), y(0), z(0)) = (0.5, -0.5, -0.5)$.

*Author contributions.* T.M. conceived the study and conducted the analyses with contributions from N.B. Both authors discussed and interpreted results. T.M. wrote the manuscript with contributions from N.B.

*Competing interests.* The authors declare that they have no competing financial interests.

*Acknowledgements.* The authors thank Keno Riechers and Maya Ben-Yami for their helpful comments. The authors acknowledge funding by

the Volkswagen Foundation. This is TiPES contribution #X; The TiPES ("Tipping Points in the Earth System") project has received funding from the European Union's Horizon 2020 research and innovation programme under grant agreement No 820970. N.B. acknowledges further funding by the European Union's Horizon 2020 research and innovation programme under the Marie Sklodowska-Curie grant agreement No. 956170, as well as from the Federal Ministry of Education and Research under grant No. 01LS2001A.

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

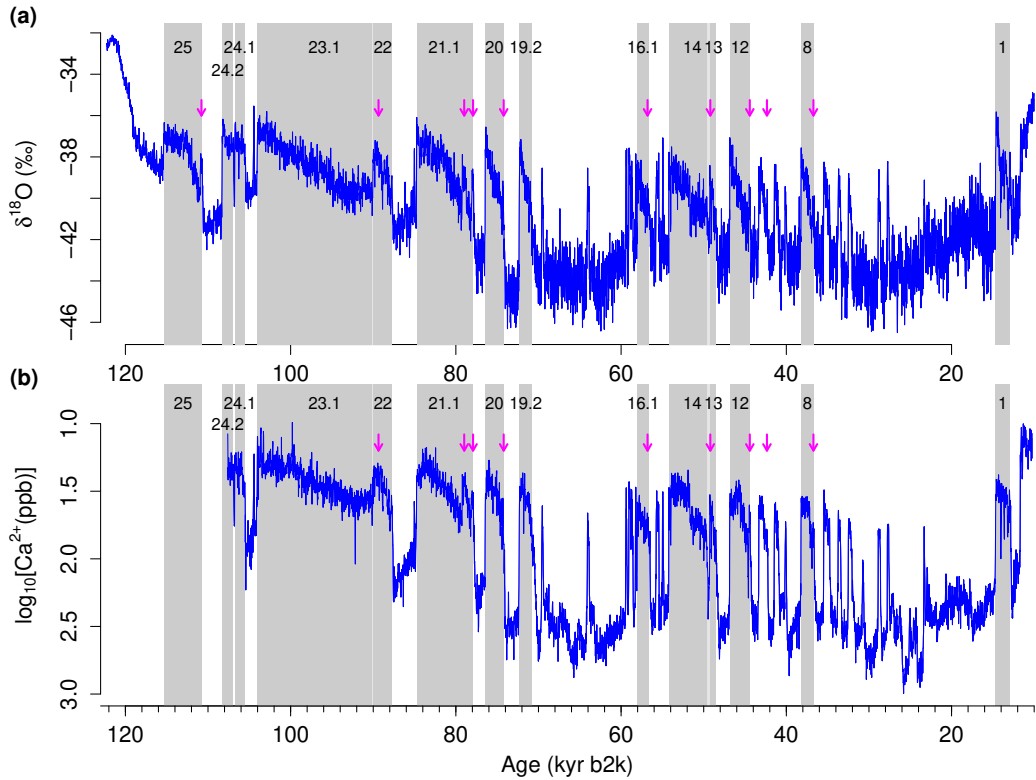

**Figure 1.** Greenland records from the NGRIP ice core: (a) $\delta^{18}$O and (b) $\log_{10}[Ca^{2+}]$ (Seierstad et al., 2014; Rasmussen et al., 2014). The interstadial parts longer than 1000-yr are highlighted with grey shades; their numbering is given at the top of each record (Rasmussen et al., 2014). The rebound events are indicated by arrows (see Section 2.1 for their list). Both records are presented at 20-yr resolution. The $\log_{10}[Ca^{2+}]$ record is available only up to DO-24.1. The compositions of GI-23.1 and GI-22, as well as of GI-14 and GI-13, are considered individual long interstadials (Capron et al., 2010; Rasmussen et al., 2014). The vertical axis for $\log_{10}[Ca^{2+}]$ in (b) is reversed to ease visual comparison with the $\delta^{18}$O record.

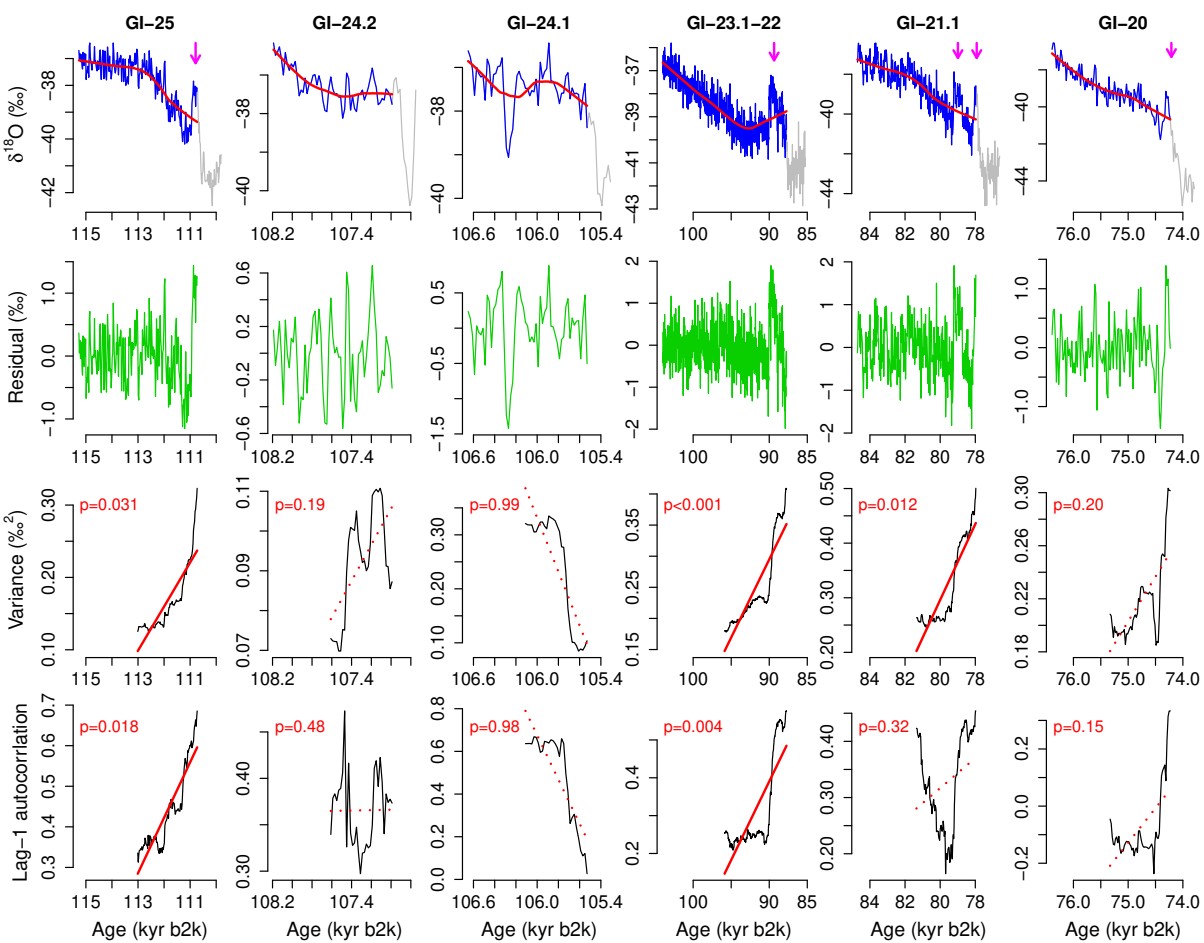

**Figure 2.** Analysis of CSD-based precursor signals of abrupt DO cooling transitions, for the first 6 interstadials of NGRIP $\delta^{18}$O, from 115 ka to 74 kyr b2k. (Top row) Interstadials longer than 1000 yr (blue). The cooling transition and stadial parts are shown in grey (Rasmussen et al., 2014). Nonlinear trends are calculated with the Locally Weighted Scatterplot Smoothing (LOESS) (red). The smoothing span $\alpha$ that defines the fraction of data points involved in the local regression is set to 50% of each interstadial length. The rebound events are indicated by arrows (see Section 2.1). (Second row) Residuals (green) resulting from subtracting the nonlinear trends (red) from the records (blue). (Third row) Variance estimate in rolling windows (black) with size equal to 50% of each interstadial length. Values are plotted at the right edge of each rolling window. The linear trend is shown by a solid red line for $p < 0.05$, by a dashed red line for $0.05 < p < 0.1$, and by a dotted line for $p > 0.1$. (Fourth row) Same as third row but for the lag-1 autocorrelation (i.e., a lag of 20 yr).

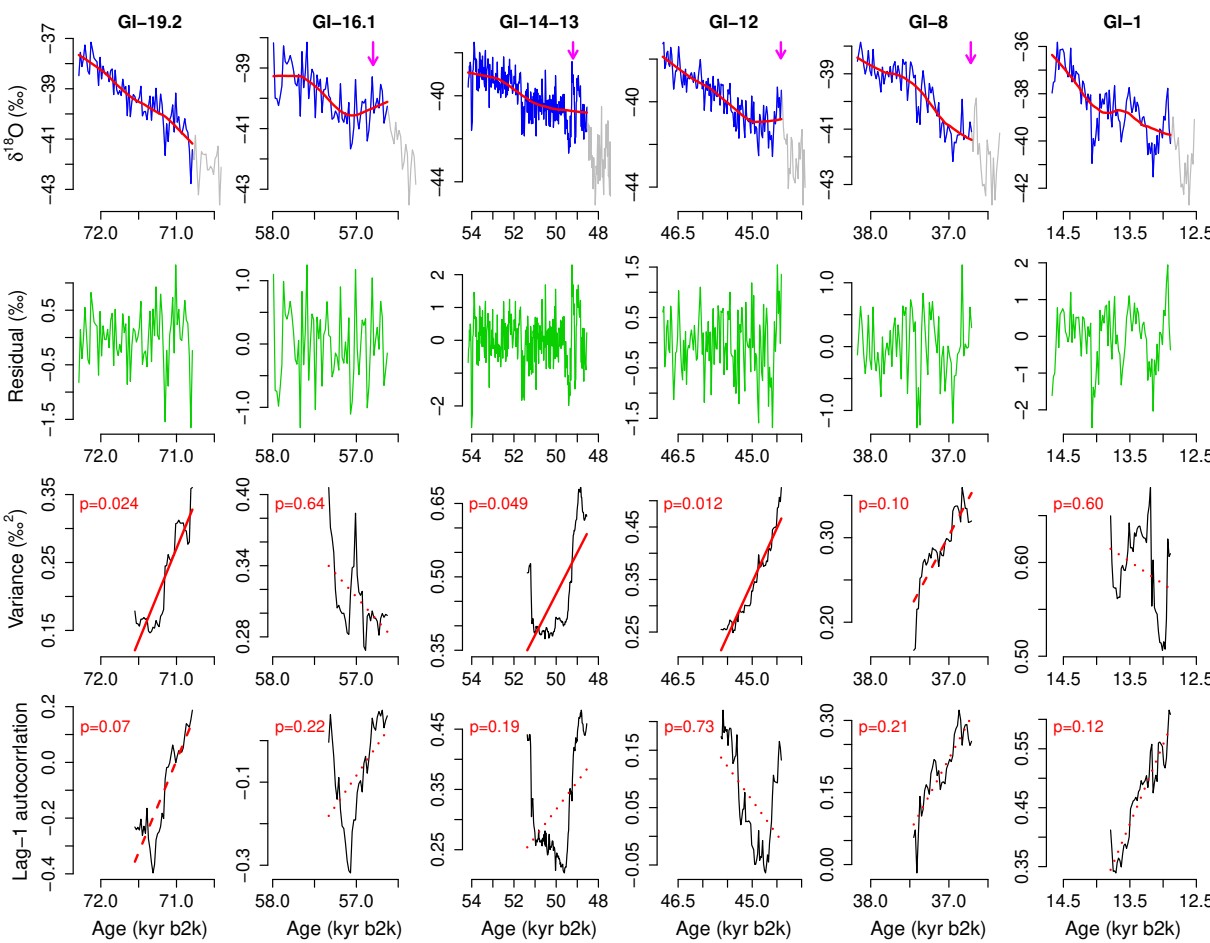

**Figure 3.** Same as Fig. 2 but for the following 6 interstadials, from 74 ka to 12 kyr b2k.

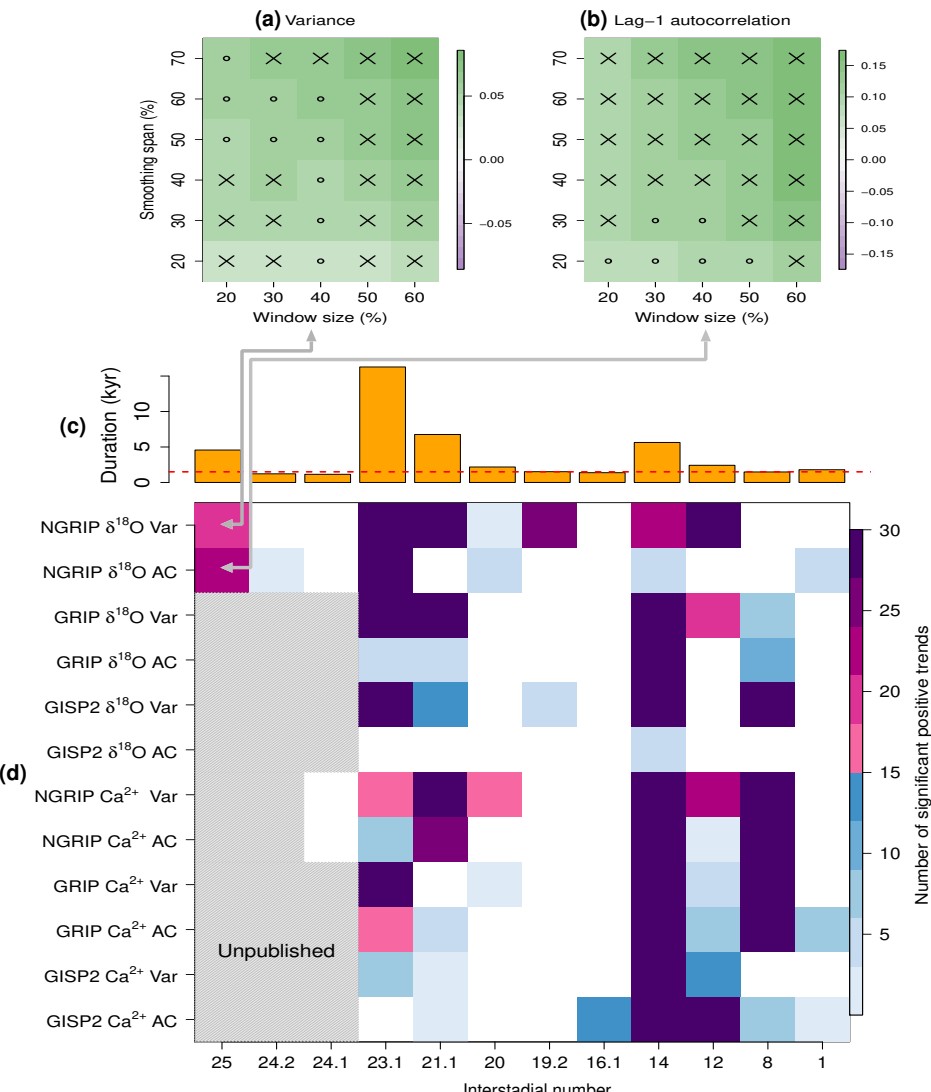

**Figure 4.** Detection of precursor signals of DO cooling transitions for different interglacials, different proxy variables, different ice cores, and different CSD indicators. (a,b) Robustness analysis of precursor signals with respect to the smoothing span and the rolling window size (% of interstadial length): the case of GI-25 interstadial from the NGRIP $\delta^{18}$O record. The CSD indicator is the variance in (a) and the lag-1 autocorrelation in (b). Cross marks (x) indicates statistically significant positive trend of the respective CSD indicator ($p < 0.05$) based on a phase surrogate test (see Section 2.2), small open circles (o) indicate barely significant positive trends ($0.05 < p < 0.1$) and cells are left blank if $p > 0.1$. (c) Durations of interstadials longer than 1000 yr (Table S1). The dashed line indicates 1500 yr (d) Robustness of finding precursor signals for DO cooling transitions. The color indicates the number significant ($p < 0.05$) positive trends in each of the 30 sets of the smoothing spans and the rolling window sizes as in (a) and (b). For the cases of grey-shaded cells, the data is not publicly available.

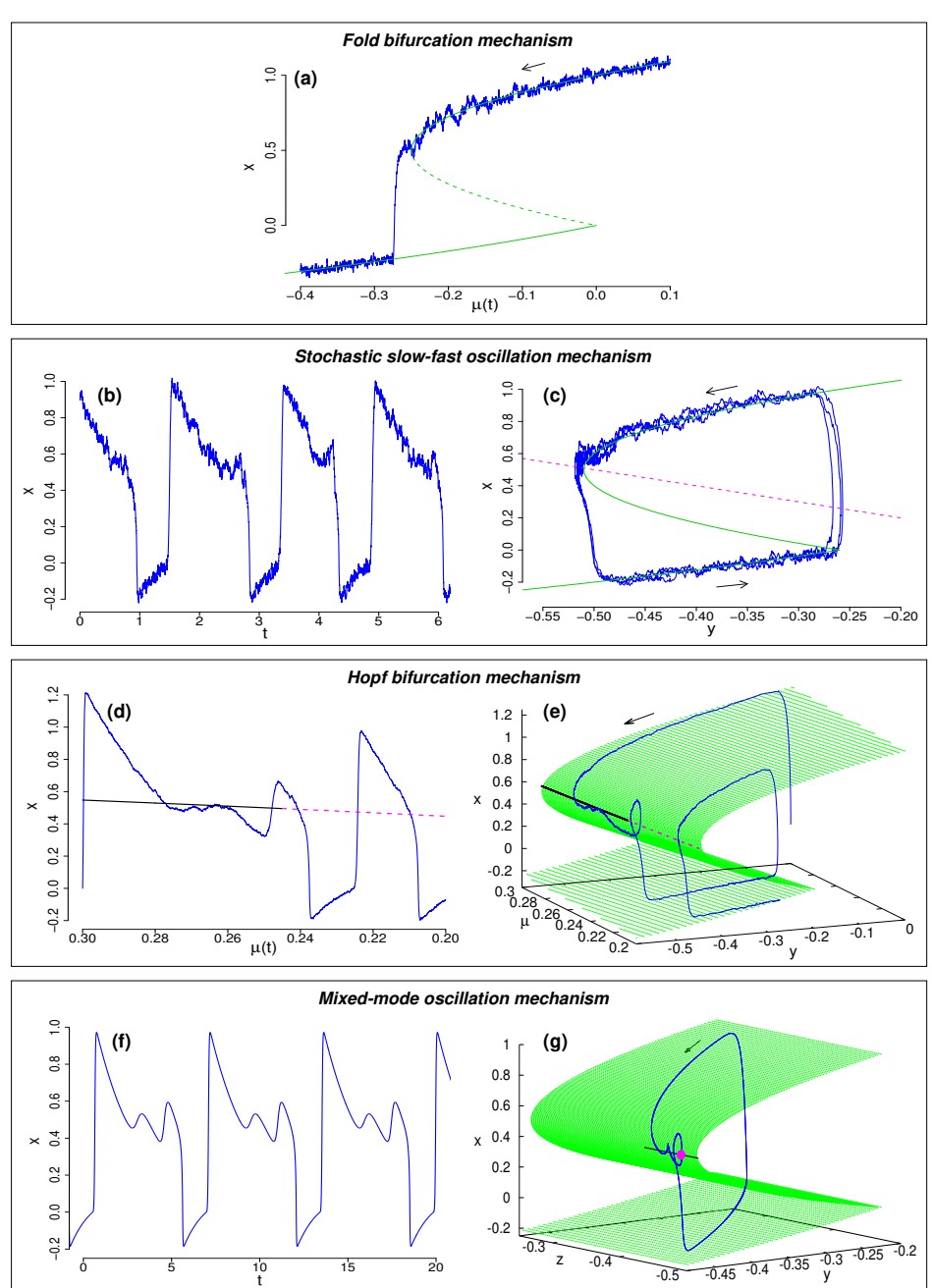

**Figure 5.** Four potential dynamical mechanisms for the DO cooling transitions: (a) Fold bifurcation mechanism. The time series $x(t)$ for decreasing $\mu(t)$ (blue). The green lines show the stable (solid) and unstable (dashed) fixed points. (b,c) Stochastic slow-fast oscillation mechanism of a FHN-type model. An example time series $x(t)$ is shown in (b) and the phase space trajectory (blue) in (c); the $x$-nullcline, i.e., the critical manifold, is shown in green. The $y$-nullcline is shown in dashed magenta. (d,e) Hopf bifurcation mechanism. An example time series $x(t)$ is shown in (d) as a function of $\mu(t)$. Stable (black, solid) and unstable (magenta, dashed) fixed points are also shown. The corresponding phase space trajectory $(x(t), y(t))$ for decreasing $\mu$ is shown in (e) in blue. The critical manifold (green). (f,g) Mixed-mode oscillation mechanism. An example time series $x(t)$ is shown in (f) and the corresponding phase space trajectory in (g). The magenta dot is the saddle point with a stable manifold in the direction of the black segment; the trajectory is spiralling around it.

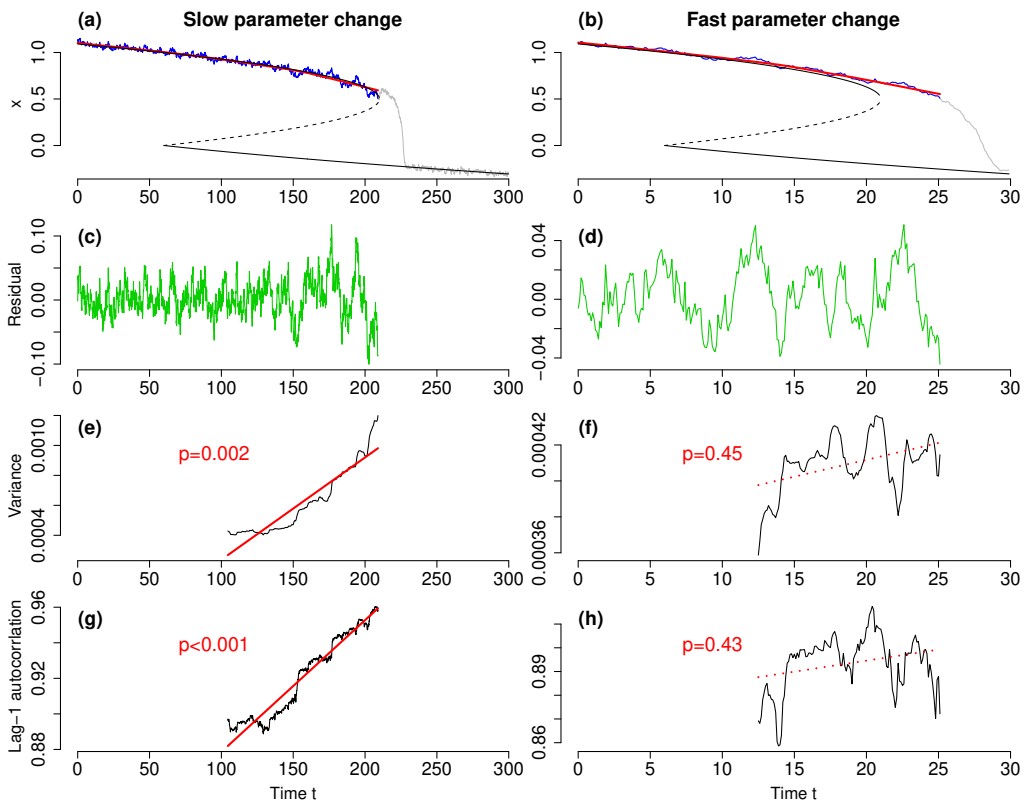

**Figure 6.** Rate-dependence of CSD indicators for the fold bifurcation in the Stommel model. Its parameter $\mu(t)$ is decreased from $\mu = 0.1$ to $\mu = -0.4$ across the fold bifurcation point $\mu = -0.25$ with a small rate $\dot{\mu}(t) = -1/600$ (left column) or with a 10 times larger rate $\dot{\mu}(t) = -1/60$ (right column). (a, b) Dynamical variable $x(t)$ before the tipping (blue) and after the tipping (gray). The time interval of sampling $x(t)$ value is 0.1 in both cases. The black solid lines and the dashed line show the stable equilibria and the unstable equilibrium, respectively. (c, d) Residuals (green) resulting from subtracting the nonlinear trends (red) from the records (blue). (e, f) Variance estimate in rolling windows (black) with size equal to 50% of the signal before tipping. Values are plotted at the right edge of each rolling window. The linear trend is shown by a solid red line for $p < 0.05$ and by a dotted line for $p > 0.1$. (g, h) Same as third row but for the lag-1 autocorrelation. Significant increases of CSD-indicators are unlikely to be observed when the parameter changes rapidly. Here typical cases are shown. Details of trajectories and the changes of CSD-indicators depend on noise realizations.