# Peer review of "Statistical precursor signals for Dansgaard-Oeschger cooling transitions"

_EGUsphere, 2023_

## Author Comment (AC1)

**Reply to Referee 1**

First of all, thank you very much for reviewing our manuscript in detail and giving us very valuable feedback. In what follows, we respond to your comments and questions, point by point, and propose changes to the manuscript in accordance. We think that these changes will improve the quality and clarity of our manuscript.

In order to improve the readability of our replies we applied a color/type coding to discriminate our replies from the referee's comments. We have attached our replies as a pdf document since color coding is not available in the browser based text editor.

Color/type coding:
*Comment by the referee.*
Reply from the authors.

*While I think the paper can be published as it is, I suggest a few minor points that could improve the presentation of the results:*

*1. The authors have used Dansgaard-Oeschger (DO) events as indicators of tipping in the AMOC. They briefly mention this in the abstract (line 5) and later provide some references in the introduction (lines 48-51). However, if the main message of the paper is to propose "robust precursor signals for a possible future AMOC collapse," I think further discussion is required to establish a clear connection between DO events and the transition from a strong to a weak AMOC. I think the paper could benefit from a new section that addresses this point.*

Thank you for pointing out this. There are some pieces of evidence that DO events have associated reorganizations of AMOC. One is concurring changes of the North Atlantic temperatures and the ocean circulation indices such as Pa/Th (e.g., Henry et al., Science 2016). Recent general circulation models also support that the changes in AMOC and its meridional heat transport are key elements of DO oscillations, as briefly mentioned in the introduction (line 48-51). Thus we suppose that DO cooling transitions recorded in Greenland records reflect past AMOC tippings. In the revised manuscript, we will add explanations about the connection between DO events and AMOC changes, to thoroughly address the referee's concern.

Our results show that several DO cooling transitions are indeed preceded by statistical precursor signals. This may increase our confidence that an AMOC transition from the strong to the weak state can be captured by the critical-slowing-down-based statistical precursor signals (e.g., Boulton et al. 2014; Boers, 2021). However, we have not proposed that the same precursor signals 'must' be observed at a possible future AMOC collapse since from the mechanistic point of view the recent AMOC weakening, which is likely driven by global warming, is different from the past AMOC declines during the glacial period (as already mentioned in line 62). We will emphasize this more in the revised manuscript.

Ref. Henry, L. G., et al. "North Atlantic ocean circulation and abrupt climate change during the last glaciation." Science 353.6298 (2016): 470-474.

*2. It is well established that changes in variance and autocorrelations are good indicators of critical slowing down (occurring during codimension one bifurcations). However, does this approach work as effectively for more complex tipping mechanisms, such as excitability (suggested in section 4 as a possible mechanism)?*

Thank you for this remark. We consider that there exist chances to observe statistical precursor signals (SPS) in the critical slowing down indicators if the fast subsystem has a critical point (like a saddle-node bifurcation point) and if a component of the slow subsystem works like a slowly-changing parameter crossing the critical point, as shown in examples in Figs. 5b-5g. However, these are not always rigorous critical slowing downs. In the example of an excitable system (Figs. 5b-5c), the underlying system always has a weakly stable fixed point, and no true bifurcation leading to critical slowing down occurs. In fact, the actual tipping in this case would be noise-induced. However, we can effectively observe the SPS in the critical slowing down indicators in this case as well, since the system would in each cyclic iteration move from more stable to less stable conditions until it finally tips to initiate the next cycle; and this partial decrease in stability is imprinted in the CSD indicators (Fig. S29). Definitely each high-dimensional mechanism giving rise to SPS in Section 4 must be investigated in more detail. We will mention this in the revised manuscript and will suggest further theoretical work in this regard as a valuable topic for further research.

*3. I would like to draw attention to the rate-induced mechanism, where an excessively rapid change in forcing can tip the system even before reaching the bifurcation point. This mechanism could arise from mechanism 3 (the Hopf bifurcation), where the system can cross the unstable limit cycle (regular threshold) and tip. It could also be relevant to mechanism 4, where the rate of forcing might push the system to cross an irregular threshold in the form of a maximal canard. Please see (Wieczorek et al. 2023) and (O'Sullivan et al. 2023), for more details:*

*Wieczorek, Sebastian, Chun Xie, and Peter Ashwin. "Rate-induced tipping: Thresholds, edge states, and connecting orbits." Nonlinearity 36.6 (2023): 3238.*

*O'Sullivan, Eoin, Kieran Mulchrone, and Sebastian Wieczorek. "Rate-induced tipping to metastable zombie fires." Proceedings of the Royal Society A 479.2275 (2023): 20220647.*

Thank you for pointing out the rate-induced mechanism and providing these useful references. Indeed the rate-induced tipping is proposed as a possible mechanism of AMOC shutdown especially under a rapid increase in freshwater forcing (e.g., Alkhayuon et al. 2019; Lohmann and Ditlevsen 2021; Ritchie et al. 2023). While we have interpreted DO cooling transitions as an analogue of bifurcation-induced tipping (with slowly changing parameter), the rate-induced mechanism (with rapidly changing parameter) is definitely worth mentioning. We will mention it in the revised manuscript.

---

## Author Comment (AC2)

**Reply to Referee 2**

First of all, thank you very much for reviewing our manuscript in detail and giving us very valuable feedback. In what follows, we respond to your comments and questions, point by point, and propose changes to the manuscript in accordance. We think that these changes will improve the quality and clarity of our manuscript.

In order to improve the readability of our replies we applied a color/type coding to discriminate our replies from the referee's comments. We have attached our replies as a pdf document since color coding is not available in the browser-based text editor.

Color/type coding:
*Comment by the referee.*
Reply from the authors.

*I was sad to see the code used in this paper was not shared 'by default'.*

We are sorry for that. We will make our codes available by default in the revised manuscript. Generally, we prefer to publish code only upon acceptance of the manuscript but of course we are happy to share the code with the referees already beforehand.

*There are lots of time series here: 2 proxy variables, 2 early warning signals, 3 cores, 12 interstadials, 5 window sizes and 6 smoothing spans leading to 3480 analysed time series (when accounting for the fact that not all cores have all interstadials in them), although relatively few give SPS (31%). If there is a common mechanism at work, why is this the case?*

As mentioned in the manuscript, robust statistical precursor signals (SPS) are more likely to be observed in longer interstadials than in shorter ones (cf. Figs 4c and 4d). We speculate that it happens because, for short interstadials, it is difficult to detect statistical changes in the stability. In the revised manuscript, we provide such a numerical example as a supplementary material and will expand the discussion section to make this clearer.

*Can the authors be sure the results are not down to chance? The authors argue that more SPS are observed than would be expected by chance, but none of the time series for each interstadial are truly independent, and there being a false positive in one time series would increase the chance of there being a false positive in another.*

As you point out, the time series of different proxies and cores are not independent if they are observations from the same DO event. Thus, whether the significant SPS are obtained by chance or not should be assessed proxy by proxy and core by core, that is, row by row in Fig. 4d. Even if we assess the results in this way, the risk of obtaining the results by chance is quite low.  For example, for the case of the variance of NGRIP d18O, we observed 6 robust SPS over 12 interstadials, which is extremely unlikely to  be obtained by chance. We have obtained no robust SPS for the case of the autocorrelation of GISP2 d18O. This is the only case where we cannot reject the null hypothesis.

*Furthermore, looking at figure 4d, different cores give different results for the same interstadial, e.g. in GI-20 only 4 cores give any SPS and only one 'robustly'. Different interstadials give different amounts of SPS, for example GI-14 gives many robust SPS but GI-19.2 doesn't. How do the authors account for this?*

Thank you for raising this question. Whether the robust SPS is observed or not depends on the proxy and core. This is possibly because different proxies from different cores are contaminated by different types and magnitudes of noise. For example, d18O may record local temperature fluctuations and Ca2+ may record turbulent fluctuations of local wind circulations. We will comment on this point in the revised manuscript.

*When looking for SPS, the time series must be decomposed into an slowly changing equilibrium state and fluctuations about that state. As a lot of the signal in this case for SPS comes from 'rebound events' the authors are assuming that the rebound events represent fluctuations rather than changes in the equilibrium state. What is the justification for this?*

We assume that the rebound events present fluctuations due to loss of stability of quasi-equilibrium states, given that the four low-dimensional dynamical systems in Fig. 5 can qualitatively mimic the rebound events. The alternative assumption of rebound events as intermediate equilibrium states is mentioned in the discussion section, citing Lohmann et al. The justification of our assumption is beyond the scope of this work. We will more explicitly note this assumption in the revised manuscript.

*The authors may want to discuss mechanisms that can lead to changes in variance and autocorrelation not due to changing stability but due other factors. For example, due to changes in the properties of the climate forcing.*

Thank you for this comment. In the revised manuscript, we will mention that the increases in variance toward a DO cooling transition may be explained by

the larger climatic fluctuations in colder states, but the increases in autocorrelation cannot generally be explained by it. We will nevertheless add a discussion on the possibility of false positive and negative SPS.

*Furthermore, changes in the statistical properties in the measurement process may also affect the results. For example, measurements in the ice cores further in the past may be more uncertain and therefore noisier, but measurements closer to the present may be less noisy and therefore more correlated.*

We fully agree that the uncertainty of the data is higher in the older part of the cores. On the other hand, we don't see systematic changes in the results detecting SPS (e.g., Fig. 4d).

*Specific Comments:*

*Line 11: Should be rebound events not rebound event*

Corrected.

*Line 15: This tipping definition excludes N-tipping, which has no thresholds. Different authors define tipping differently but as there is disagreement over whether DO events are N or B tipping I wonder if it is better to adopt a definition compatible with the Ashwin 2012 typology?*

Thank you for this comment. Yes, the description of "tipping points" in line 15 was restricted to that of bifurcation-induced tipping points. We will adopt a wider description in the first paragraph such as "a threshold crossed irreversibly by the system's dynamics". Then in the second paragraph, we will mention B-, N-, and R-tipping citing Ashwin et al. and others.

*Line 91 "R^2 = 0.95", what fit is this measuring?*

R^2 = 0.95 is the coefficient of determination for the correlation between the length of rebound event and the length of interstadial. In the revised manuscript, we have rewritten this and now say that their durations are correlated with R^2=0.95.

*Line 114: The autocorrelation is different to that in Bury et al who have C(tau) = (cos (omega tau)) exp(mu |tau|)*

Thank you. Corrected!

*Line 115: Should be "increase or decrease"*

Corrected.

*Line 117: How do the authors know tau is sufficiently small, especially as omega may also be changing?*

In theory, we can calculate the autocorrelation function over the running window and thus can choose a sufficiently small tau. Here, the minimum sampling time is taken. The frequency omega itself does not change across the Hopf bifurcation (Strogatz 2018).

*Line 120: Is a linear fit suitable if half of the interstadial is used i.e. 500+ years? Could the stable state be changing nonlinearly in this period?*

The locally estimated scatterplot smoothing (LOESS) used in this study performs a local polynomial (here simply linear) fit in its procedure, giving more weight to points near the point whose response is being estimated and less weight to points further away. Thus, it can provide smoothed series for time series with nonlinear trend even if the local fit is linear. We will explain this point in detail in the revised manuscript.

*Line 187: makes reference to interstadials shorter than 1000 years but Line 106 implies the authors are excluding interstadials shorter than 1000 years. Have I misunderstood?*

Sorry the sentence was misleading. In section 3.1, we have analyzed interstadials longer than 1000 years, but we have examined high-resolution interstadial data longer than 300 years in section 3.2. Thus the data between 300 and 1000 years is actually included in section 3.2. In order to avoid confusion, we simply say that "Robust SPS have not been observed for short interstadials again."

*Line 295-298: "can be shown to be 0.05". I think it would be helpful to show this. When I run the authors code I do not get any output like 0.044 or 0.042, but I may be running the code incorrectly. Is this calculation included in the shared code?*

Excuse me, we have included the codes for main figures, but some of the codes for appendices and supplementary files are not included because some of them are tedious. However, we will upload all of the codes to a repository when submitting the revision.

*Figure 4d: Could the colormap used in this figure be changed to a diverging colormap, with its centre at 15, so that it is easy to see if an SPS is robust. Currently*

*it is difficult to know if the colours correspond to values larger than or smaller than 15.*

Thank you for this comment. We will improve the Figure 4d so that we can know if the corresponding values are above or below 15.

---

## Author Response (AR1)

Dr. Takahito Mitsui
Earth System Modelling, School of Engineering & Design
Technical University of Munich
80333 Munich, Germany
takahito321@gmail.com

17 December 2023

Dear Prof. Risebrobakken,

Thank you very much for reviewing our manuscript entitled "Statistical precursor signals for Dansgaard-Oeschger cooling transitions". We herewith resubmit a new version of our manuscript, which has been revised following the referees' comments. You'll find a list of the main corrections and our point-by-point responses to the comments below. We think that these changes in response to the points raised by the referees have substantially improved the quality and clarity of our manuscript.

**Major changes in the revised manuscript**

- Following comments by the editor and reviewer 1, we have expanded the description (4th paragraph in Section 1) about the DO events and its possible connection to the AMOC changes. On the other hand, given the still elusive link between the DO events and the past AMOC changes, we have stated that "if the DO events reflect the past AMOC changes, our result on the predictability of DO events may have an implication on the possible future AMOC collapse."

- Reviewer 2 asked why there are quite several DO cooling transitions without showing significant statistical precursor signals (SPS). We now discuss possible reasons in Section 4, where we include a **new Figure 6** in order to illustrate that the detectability of the statistical precursor signals can depend on the rate of the parameter change. Also in the revised manuscript, we more explicitly mention that robust SPS are observed in most interstadials longer than roughly 1500 years but not in short interstadials, in line with our conceptual models.

- We have made a **new Table S1**, which shows the lengths of interstadials analyzed, and the previous Table S1 is now Table S2.

- Following the suggestion of Reviewer 2, we have changed the color scheme of Fig. 4d and S28b so that it is clear whether the number of detected significant precursor signals is below or above the threshold 15.

- We have mentioned that the code used for this study is available from https://github.com/takahito321/Predictability-of-DO-cooling.git

**Point-by-point reply to the reviewers' comments**

In order to improve the readability of our replies we applied a color/type coding to discriminate our replies from the referee's comments.

Color/type coding:
*Comments by the reviewers and public comment.*
Reply from the authors.

**Reply to Editor's comment.**

*Related to the comment on AMOC and AMOC collapse, made by reviewer 1, and the need to add to the discussion, I will recommend that you also acknowledge the ongoing discussion with respect to the potential role of AMOC changes as a driver behind DO-events, or not. As you mention, there are studies arguing for AMOC changes to be a critical factor for DO-events, however, there are also studies e.g., stressing the importance of the gyre component rather than AMOC (e.g. Li and Born, 2019). You do mention in the introduction that the mechanism behind DO-events is debated, but it may be good to consider expanding a bit on what the other suggested mechanisms are and what your reasoning are for focusing on the potential role of AMOC.*

Thank you for this valuable comment. In the revised manuscript, we have expanded the paragraph about the DO events and their possible relation to the AMOC. Please see the fourth paragraph of the Introduction. As suggested, it includes non-AMOC-centric views of DO events (e.g., Li and Born 2019; Dokken et al., 2013). Given the still elusive link between the DO events and the past AMOC changes, we articulate such that _if_ the DO events reflect the past AMOC changes, our result on the predictability of DO events may have an implication on the possible future AMOC collapse.

**Reply to Reviewer 1**

*1. The authors have used Dansgaard-Oeschger (DO) events as indicators of tipping in the AMOC. They briefly mention this in the abstract (line 5) and later provide some references in the introduction (lines 48-51). However, if the main message of the paper is to propose "robust precursor signals for a possible future AMOC collapse," I think further discussion is required to establish a clear connection between DO events and the transition from a strong to a weak AMOC. I think the paper could benefit from a new section that addresses this point.*

Thank you for this valuable comment. As mentioned in the reply to the Editor, the implication for the possible future AMOC collapse is not the main message of this article. Our focus is on investigating the predictability of past abrupt climate changes focussing on the example of DO cooling events, from the viewpoint of tipping point theory.

If any, we agree that we should establish a clear connection between DO events and the AMOC transition. As we wrote in the reply to the Editor, we have expanded the paragraph about the DO events and their possible relation to the AMOC, including also non-AMOC-

centric views of DO events (e.g., Li and Born 2019; Dokken et al., 2013). See the fourth paragraph in the Introduction. Given that the relation between the DO events and the AMOC changes is still not fully established, we discuss such that if (and only if) the DO events reflect the past AMOC changes, our result on the predictability of DO events may have an implication on the possible future AMOC collapse.

*2. It is well established that changes in variance and autocorrelations are good indicators of critical slowing down (occurring during codimension one bifurcations). However, does this approach work as effectively for more complex tipping mechanisms, such as excitability (suggested in section 4 as a possible mechanism)?*

We consider that there exist chances to observe statistical precursor signals (SPS) in the critical slowing down indicators if the fast subsystem has a critical point (like a saddle-node bifurcation point) and if a component of the slow subsystem works like a slowly-changing parameter approaching the critical point, as shown in examples in Figs. 5b-5g. The example of an excitable system (Figs. 5b-5c) is however not a rigorous bifurcation-induced tipping; the underlying system always has a weakly stable fixed point, and no true bifurcation leading to critical slowing down occurs. In fact, the actual tipping in this case would be noise-induced. However, we can effectively observe the SPS in the critical slowing down indicators in this case as well, since the system would in each cyclic iteration move from more stable to less stable conditions until it finally tips to initiate the next cycle; and this partial decrease in stability is imprinted in the CSD indicators (Fig. S29). Definitely, each high-dimensional mechanism giving rise to SPS in Section 4 must be investigated in more detail. We have mentioned this point in the discussion of the revised manuscript: "The SPS for bifurcation-induced tippings including cases (1) and (3) above are established. However, detailed characterizations of SPS for the stochastic slow-fast oscillations of the excitable system (case 2) and for the mixed-mode oscillations (case 4) remain to be elucidated."

*3. I would like to draw attention to the rate-induced mechanism, where an excessively rapid change in forcing can tip the system even before reaching the bifurcation point. This mechanism could arise from mechanism 3 (the Hopf bifurcation), where the system can cross the unstable limit cycle (regular threshold) and tip. It could also be relevant to mechanism 4, where the rate of forcing might push the system to cross an irregular threshold in the form of a maximal canard. Please see (Wieczorek et al. 2023) and (O'Sullivan et al. 2023), for more details:*

*Wieczorek, Sebastian, Chun Xie, and Peter Ashwin. "Rate-induced tipping: Thresholds, edge states, and connecting orbits." Nonlinearity 36.6 (2023): 3238.*

*O'Sullivan, Eoin, Kieran Mulchrone, and Sebastian Wieczorek. "Rate-induced tipping to metastable zombie fires." Proceedings of the Royal Society A 479.2275 (2023): 20220647.*

Thank you for pointing out the rate-induced tipping mechanism and providing these useful references. Indeed, rate-induced tipping has been proposed as a possible mechanism of AMOC shutdown especially under a rapid increase in freshwater forcing, for which we have cited Alkhayuon et al. 2019, Lohmann and Ditlevsen 2021, and Ritchie et al. 2023 in the revised manuscript. While we have interpreted DO cooling transitions as an analogue of bifurcation-induced tipping (with slowly changing parameter), the rate-induced mechanism

(with rapidly changing parameter) is definitely worth mentioning. In the revised manuscript, we have mentioned the rate-induced mechanism in Section 1 (second paragraph) citing the above references suggested by the reviewer. Also in Section 5 (third paragraph), we have mentioned that "The lack of observed SPS for the interstadials less than roughly 1500 years shows a rate-dependent aspect of DO cooling transitions. However, a comprehensive investigation of DO cooling transitions from the viewpoint of rate-induced tipping is beyond the scope of this work."

**Reply to Reviewer 2**

*I was sad to see the code used in this paper was not shared 'by default'.*

We are sorry for that. We will make our codes available by default in the revised manuscript. Generally, we prefer to publish code only upon acceptance of the manuscript but of course we are happy to share the code with the referees already beforehand. See  https://github.com/takahito321/Predictability-of-DO-cooling.git.

*There are lots of time series here: 2 proxy variables, 2 early warning signals, 3 cores, 12 interstadials, 5 window sizes and 6 smoothing spans leading to 3480 analysed time series (when accounting for the fact that not all cores have all interstadials in them), although relatively few give SPS (31%). If there is a common mechanism at work, why is this the case?*

We can consider several causes for the relatively low fraction (31%) of observed robust statistical precursor signals (SPS). First, each proxy and core can be contaminated by different types/magnitudes of noise (e.g., d18O may record local fluctuations of temperatures and log Ca2+ turbulent fluctuations of local wind circulations). Second, the SPS are not observed for interstadials shorter than roughly 1500 years. The SPS could not be observed possibly because the time-scale-separation assumption for the critical slowing down is violated for the short interstadials. This possibility is for example demonstrated with the Stommel-type model in the new Fig. 6. Third, in several cases, the critical-slowing-down (CSD) indicators decrease first and then increase. The first decrease, harming the monotonic increases of CSD indicators, might reflect a memory of the preceding DO warming transition. In real world applications, the significance level at 0.05 is considered conservative, and 0.1 is sometimes chosen (Laitinen et al., Ecol Evol. 2021; Dakos et al. 2012). Therefore, we consider the number of detected SPS in this work not few given the noisy empirical proxy records; in particular, the number of robust SPS is much higher than expected to arise by chance.  The above three points are mentioned in the revised manuscript.

*Can the authors be sure the results are not down to chance? The authors argue that more SPS are observed than would be expected by chance, but none of the time series for each interstadial are truly independent, and there being a false positive in one time series would increase the chance of there being a false positive in another.*

As you point out, the time series of different proxies and cores are not independent if they are observations from the same DO event. Thus, whether the significant SPS are obtained

by chance or not should be assessed proxy by proxy and core by core, that is, row by row in Fig. 4d. Even if we assess the results in this way, the risk of obtaining the results by chance is quite low. For example, the probability for obtaining more than 3 robust SPS from 12 independent stationary samples is only 2%. For the case of the variance of NGRIP d18O, we observed 6 robust SPS over 12 interstadials, which is extremely unlikely to be obtained by chance. In the revised manuscript, this is mentioned in Section 3.1 (the last paragraph).

*Furthermore, looking at figure 4d, different cores give different results for the same interstadial, e.g. in GI-20 only 4 cores give any SPS and only one 'robustly'. Different interstadials give different amounts of SPS, for example GI-14 gives many robust SPS but GI-19.2 doesn't. How do the authors account for this?*

Thank you for raising this question. Whether a robust SPS is observed or not depends on the proxy and core. This is possibly because different proxies from different cores are contaminated by different types and magnitudes of noise. For example, d18O may record local temperature fluctuations and Ca2+ may record turbulent fluctuations of local wind circulations. In the revised manuscript, this has been commented on in Section 3.1 (the last paragraph).

*When looking for SPS, the time series must be decomposed into a slowly changing equilibrium state and fluctuations about that state. As a lot of the signal in this case for SPS comes from 'rebound events' the authors are assuming that the rebound events represent fluctuations rather than changes in the equilibrium state. What is the justification for this?*

We assume that the rebound events present fluctuations due to loss of stability of quasi-equilibrium states, given that the four low-dimensional dynamical systems in Fig. 5 can qualitatively mimic the rebound events. The alternative assumption of rebound events as intermediate equilibrium states is mentioned in the discussion section, citing Lohmann et al. (preprint). The justification of our assumption is beyond the scope of this work and left for a future study. In the revised manuscript, this is briefly mentioned in Section 4 (third paragraph).

*The authors may want to discuss mechanisms that can lead to changes in variance and autocorrelation not due to changing stability but due other factors. For example, due to changes in the properties of the climate forcing.*

Thank you for this comment. In the revised manuscript, we have mentioned that one might link the increase in the proxy variance with the tendency of larger climatic fluctuations in colder states (Ditlevsen, 1996), but the increases in autocorrelation cannot generally be explained by it. See Section 5 (first paragraph).

*Furthermore, changes in the statistical properties in the measurement process may also affect the results. For example, measurements in the ice cores further in the past may be more uncertain and therefore noisier, but measurements closer to the present may be less noisy and therefore more correlated.*

We fully agree that the uncertainty of the data is higher in the older part of the cores. On the other hand, we don't see systematic changes in the results detecting SPS (e.g., Fig. 4d).

*Specific Comments:*

*Line 11: Should be rebound events not rebound event*

Corrected.

*Line 15: This tipping definition excludes N-tipping, which has no thresholds. Different authors define tipping differently but as there is disagreement over whether DO events are N or B tipping I wonder if it is better to adopt a definition compatible with the Ashwin 2012 typology?*

Thank you for this comment. Yes, the description of "tipping points" in line 15 was restricted to that of bifurcation-induced tipping points. In the revised manuscript, we adopt a wider definition by IPCC 6[th] assessment report: "a critical threshold beyond which a system reorganizes, often abruptly and/or irreversibly". Here the term "threshold" implies (i) a bifurcation point in B-tipping, (ii) a basin boundary or a saddle in N-tipping, and (ii) a critical rate of parameter changes in R-tipping, respectively. Thus, it includes all the three major types of tipping points. In the second paragraph in the Introduction, we have described B-, N-, and R-tipping citing Ashwin et al. and others.

*Line 91 "R^2 = 0.95", what fit is this measuring?*

R^2 = 0.95 is the coefficient of determination for the correlation between the length of rebound event and the length of interstadial. In the revised manuscript, we have rewritten this and now say that their durations are correlated with R^2=0.95.

*Line 114: The autocorrelation is different to that in Bury et al who have C(tau) = (cos (omega tau)) exp(mu |tau|)*

Thank you. Corrected!

*Line 115: Should be "increase or decrease"*

Corrected.

*Line 117: How do the authors know tau is sufficiently small, especially as omega may also be changing?*

First of all, the frequency omega itself does not change across the Hopf bifurcation (Strogatz 2018)). Since we can calculate the autocorrelation function over the running window, we can, in principle, choose a sufficiently small tau, which makes the autocorrelation positive, i.e., tau<π/(2omega). But practically we use the lag-1 autocorrlation corresponding to minimal tau. We have mentioned this point in the revised manuscript (first paragraph in Section 2.2).

*Line 120: Is a linear fit suitable if half of the interstadial is used i.e. 500+ years? Could the stable state be changing nonlinearly in this period?*

The locally estimated scatterplot smoothing (LOESS) used in this study performs a local polynomial (here simply linear) fit in its procedure, giving more weight to points near the point whose response is being estimated and less weight to points further away. Thus, it can provide smoothed series for time series with nonlinear trend even if the local fit is linear. We have explained this point in the second paragraph of Section 2.2 in the revised manuscript.

*Line 187: makes reference to interstadials shorter than 1000 years but Line 106 implies the authors are excluding interstadials shorter than 1000 years. Have I misunderstood?*

Sorry the sentence was misleading. In section 3.1, we have analyzed interstadials longer than 1000 years, but we have examined high-resolution interstadial data longer than 300 years in section 3.2. Thus the data between 300 and 1000 years is actually included in section 3.2. In the revised manuscript (last paragraph of Section 3.2), we explicitly mention that "Robust SPS have not been observed again for interstadials shorter than roughly 1500 yr (Figs S28 and 4)."

*Line 295-298: "can be shown to be 0.05". I think it would be helpful to show this. When I run the authors code I do not get any output like 0.044 or 0.042, but I may be running the code incorrectly. Is this calculation included in the shared code?*

We had included the code for the main figures, but some parts of the code for appendices and supplementary files were not included because some of them are tedious. However, now we uploaded all of the code to a repository.

*Figure 4d: Could the colormap used in this figure be changed to a diverging colormap, with its centre at 15, so that it is easy to see if an SPS is robust. Currently it is difficult to know if the colours correspond to values larger than or smaller than 15.*

Thank you for this comment. We have changed the colors of Figs. 4d and S28 so that we can know if the corresponding values are above or below 15.

---

## Author Response (AR3)

Dr. Takahito Mitsui
Earth System Modelling, School of Engineering & Design
Technical University of Munich
80333 Munich, Germany
takahito321@gmail.com

5 February 2024

Dear Prof. Risebrobakken,

Thank you very much for reviewing our manuscript entitled "Statistical precursor signals for Dansgaard-Oeschger cooling transitions". We herewith resubmit a new version of our manuscript, which has been revised following the referees' comments. You'll find our point-by-point responses to the comments below. We think that these changes in response to the points raised by the referees have improved the quality and clarity of our manuscript.

**Point-by-point reply to the reviewers' comments**

In order to improve the readability of our replies we applied a color/type coding to discriminate our replies from the referee's comments.

Color/type coding:
*Comments by the reviewers and public comment.*
Reply from the authors.

**Reply to Referee #2**

*There is a typo on line 95: "quasi-stadail" should be "quasi-stadial".*

Thank you. Corrected.

**Reply to Referee #3**

*• Line 5: "co-dimension one bifurcation" is far too technical for an abstract, and is not well defined in the manuscript. Please provide more insights for non-experts.*

In the revised manuscript, we explained that the co-dimension-1 bifurcations are, in simple terms, the bifurcations that can be typically encountered by the change of a single control parameter (Thompson and Sieber, 2011).

*• Line 5: "variance and the autocorrelation" of what? Of the time series of a variable representing the system analysed, I assume, but please clarify.*

In the revised manuscript it is written as "the variance and short-lag autocorrelations of the fluctuations increase in a stochastically forced system approaching a critical or bifurcation-induced transition".

*• Line 54: SPS is not defined here, and its definition only appears later on.*

Thank you for pointing out this. In the revised manuscript, we have noted the SPS as follows. "Thus, the changes in CSD indicators such as the increase of the variance as well as the autocorrelation can be seen as *statistical precursor signals* (SPS) of critical transitions."

*• Line 75: "with sufficient data length". Indeed, and in this respect the work of Michel et al. (2022) certainly deserves to be cited on top of Boers (2021) paper…*

Thank you for reminding us of this important work. We have cited Michel et al. (2022) in the introduction and the discussion.

*• Line 84: "20-year resolution" is quite coarse. What might be the implication of this for the utility of those data for present-day? It might be worth to discuss this caveat later on*

In the revised manuscript, we have expressed this caveat as follows " There is, however, a caveat to this implication because the past DO cooling transitions are different from the presently inferred AMOC changes. The time resolution (mainly 20 years and additionally 5 years) and the length (mainly >1000 years and additionally >300 years) of the interstadial segment data used in this study are coarser and mostly longer than the annual data used for analyzing AMOC fingerprints during the industrial period (Boers, 2021; Ben-Yami et al., 2023; Ditlevsen and Ditlevsen, 2023) and the last millennium (Michel et al., 2022) …"

*• Line 95: typo on "stadial"*

Corrected.

*• Line 121: "numerical studies" is a bit unclear. Do you mean "studies using numerical climate models"?*

Yes, these are studies analyzing the outputs of numerical climate models. In the revised manuscript, we just write "previous studies" because it is clear from the following context.

*Reference:*
*Michel S., et al. Early warning signal for a tipping point suggested by a millennial Atlantic Multidecadal Variability reconstruction. Nature Communications 13, 5176 (2022).*